# Gut Microbiota Composition Can Predict Colonization by Multidrug-Resistant Bacteria in SARS-CoV-2 Patients in Intensive Care Unit: A Pilot Study

**DOI:** 10.3390/antibiotics12030498

**Published:** 2023-03-02

**Authors:** Jorge García-García, Patricia Diez-Echave, María Eugenia Yuste, Natalia Chueca, Federico García, Jose Cabeza-Barrera, Emilio Fernández-Varón, Julio Gálvez, Manuel Colmenero, Maria Elena Rodríguez-Cabezas, Alba Rodríguez-Nogales, Rocío Morón

**Affiliations:** 1Department of Pharmacology, Center for Biomedical Research (CIBM), University of Granada, 18071 Granada, Spain; 2Instituto de Investigación Biosanitaria de Granada (ibs.GRANADA), 18012 Granada, Spain; 3Servicio de Medicina Intensiva, Hospital Universitario Clínico San Cecilio, 18016 Granada, Spain; 4Servicio de Microbiología Clínica, Hospital Universitario San Cecilio, Centro de Investigación Biomédica en Red de Enfermedades Infecciosas (CIBERInfec), 18016 Granada, Spain; 5Centro de Investigación Biomédica en Red de Enfermedades Infecciosas (CIBERInfec), 28029 Madrid, Spain; 6Servicio Farmacia Hospitalaria, Hospital Universitario Clínico San Cecilio, 18016 Granada, Spain; 7Centro de Investigación Biomédica en Red de Enfermedades Hepáticas y Digestivas (CIBERehd), 28029 Madrid, Spain

**Keywords:** Intensive Care Unit, microbiome, multidrug-resistant bacteria, SARS-CoV-2

## Abstract

The SARS-CoV-2 infection has increased the number of patients entering Intensive Care Unit (ICU) facilities and antibiotic treatments. Concurrently, the multi-drug resistant bacteria (MDRB) colonization index has risen. Considering that most of these bacteria are derived from gut microbiota, the study of its composition is essential. Additionally, SARS-CoV-2 infection may promote gut dysbiosis, suggesting an effect on microbiota composition. This pilot study aims to determine bacteria biomarkers to predict MDRB colonization risk in SARS-CoV-2 patients in ICUs. Seventeen adult patients with an ICU stay >48 h and who tested positive for SARS-CoV-2 infection were enrolled in this study. Patients were assigned to two groups according to routine MDRB colonization surveillance: non-colonized and colonized. Stool samples were collected when entering ICUs, and microbiota composition was determined through Next Generation Sequencing techniques. Gut microbiota from colonized patients presented significantly lower bacterial diversity compared with non-colonized patients (*p* < 0.05). Microbiota in colonized subjects showed higher abundance of *Anaerococcus*, *Dialister* and *Peptoniphilus*, while higher levels of *Enterococcus*, *Ochrobactrum* and *Staphylococcus* were found in non-colonized ones. Moreover, LEfSe analysis suggests an initial detection of *Dialister propionicifaciens* as a biomarker of MDRB colonization risk. This pilot study shows that gut microbiota profile can become a predictor biomarker for MDRB colonization in SARS-CoV-2 patients.

## 1. Introduction

The new coronavirus, denominated “Severe Acute Respiratory Syndrome Coronavirus 2” (SARS-CoV-2), has been the genesis of the 2019 Coronavirus Disease (COVID-19), a pandemic menace to health worldwide [1]. According to the World Health Organization (WHO), the ongoing outbreak has spread to more than 215 countries, with more than 600 million officially reported cases and over 6 million confirmed deaths [2]. Considering the symptoms, disease severity has been classified into mild, moderate, severe, and critical [3]. Even though most of the COVID-19 cases are mild to moderate, it has been estimated that 10–15% of them progress to severe, and 15–20% of these to critical, even requiring treatment in intensive care units (ICU) [4]. Critically ill patients may show evidence of respiratory failure, septic shock, and/or multiple organ dysfunction [2,3]. SARS-CoV-2 has also aggravated another pandemic, the antimicrobial resistance (AMR), also known as a «hidden pandemic». AMR has been acknowledged as one of the most serious hazards for global health, economic, and social wellness, being estimated to be the direct cause of over 1 million deaths in 2019 [5,6]. Furthermore, different studies have revealed an association between both pandemics [7,8]. Thus, it has been reported that an increment of critically ill patients have been prescribed empirical antimicrobial therapy that may not be appropriate and could also contribute to the global increase of resistant infections [9,10].

Moreover, it is well described that admission to ICUs, including in the cases of SARS-CoV-2 patients, is associated with an intestinal dysbiosis including a remarkable reduction of phylogenetic diversity in the gut microbiota [11]. This imbalanced environment permits pathogenic microorganisms such as *Clostridium difficile*, *Pseudomonas aeruginosa*, *Candida* species, vancomycin-resistant *enterococci* (VRE), and other multidrug-resistant bacteria (MDRB) to colonize the intestine [11,12,13,14,15]. Patients colonized by MDRB have worse prognoses and increased risk for septic shock, organ failure, prolonged ICU stays, and mortality [16]. The management of the gut dysbiosis in critically ill patients is nowadays considered a key factor to increase survival rates in ICU patients, and for this reason it has, nowadays, become a trending topic in clinical research. Consequently, different strategies have been implemented to prevent gut dysbiosis and therefore MDRB colonization in critically ill patients, e.g., selective digestive tract decontamination (SDD), use of probiotics, prebiotics, combinations of both (synbiotics), faecal microbiota transplantation (FMT), and microbiome-modulating agents that regulate the metabolism of microbiota [17,18,19].

Even though it has been reported that gut dysbiosis is a common feature in SARS-CoV-2-positive patients treated in the ICU [20], their specific microbiome composition could determine the risk of developing MDRB colonization. Accordingly, in-depth examination of gut microbiota composition in both groups of patients, colonized or not, could help to establish key valuable prognostic targets, thus allowing a better management of these patients. Thus, the present pilot study aims to evaluate whether the initial gut microbiota compositions of SARS-CoV-2 positive patients in ICUs could have an impact on the colonization and establishment of MDRB.

## 2. Results

### 2.1. Study Cohort Description

A total number of 17 patients with SARS-CoV-2 infection admitted to the ICU were enrolled for this pilot study (Table 1). Among them, 8 patients suffered MDRB colonization during their stay inside ICU facilities. According to the establishment and development of MDRB colonization, we divided the total number of patients into two groups, colonized and non-colonized patients. When the screening for MDRB colonization was performed, 75% of the patients were colonized by carbapenemase-producing bacteria, while the rest were colonized by *Escherichia coli* or non-fermenter bacteria (Table 2). Moreover, one patient colonized by carbapenemase-producing bacteria presented two colonizing bacteria: *E. cloacae* and *S. marcenses*.

The general clinical characteristics of non-colonized and colonized groups are shown in Table 1. Most patients enrolled were male (65%), being similarly distributed in the two groups even though most of the women were included in the non-colonized group. Regarding the comorbidities in both groups, hypertension was the most common comorbidity (41%), followed by obesity (30%). However, separately, the colonized group showed a higher incidence rate of comorbidities than non-colonized patients, the duration of stay in the ICU being similar in both groups. Additionally, two severity indexes at hospitalization were calculated, SOFA and APACHE. The results revealed no statistical differences for these indexes between both groups of patients, although a trend to show higher values in the APACHE index was observed when colonized patients were considered. On the other hand, the biochemical parameters evaluated revealed that the colonized group displayed a significant increase in LDH levels compared to non-colonized patients (*p* < 0.05). In addition, MDRB colonization also presented a tendency for promoting higher percentage of fever (63%) or antibiotic treatment (75%) at the time of ICU access, and mortality once inside ICU (38%), when compared with non-colonized patients.

### 2.2. Lower Gut Microbiota Diversity Is Associated with a High Risk of Colonization by MDRB in Critical SARS-CoV-2 Patients

To evaluate the impact of the initial gut microbiota composition in the development of MDRB colonization in SARS-CoV-2 patients, alpha and beta diversity analyses were firstly performed (Figure 1). Alpha diversity evaluation revealed that colonized patients showed a significantly lower bacterial diversity. In fact, the MDRB colonization was associated with a significant decrease in the observed species and an increment in the Inv. Simpson index (evenness) (*p* < 0.01 and *p* < 0.05 respectively), whereas no significant modification was observed when Shannon index (richness) was considered (Figure 1A).

Additionally, principal coordinate analysis (PCoA) plots for Bray–Curtis and both unweighted and weighted UniFrac distances were employed to establish the beta diversity. Accordingly, PCoA plots indicated that the two groups were very closely clustered, with no significant separation (Figure 1B–D). Therefore, the beta diversity analysis indicated that both groups of patients displayed similar structures of microbial communities.

### 2.3. SARS-CoV-2 Associated Shifts in the Initial Gut Microbiota Are Related to MDRB Colonization in ICU Facilities

The bacterial microbiota composition in critical SARS-CoV-2 patients at the time of entering the ICU showed that the most abundant phyla in all samples were *Bacteroidota*, *Bacillota*, and *Pseudomonadota*. Of note, the abundance of these phyla did not show significant differences between non-colonized and colonized patients (Figure 2A). However, when the composition at genus level was evaluated, significant differences were found between both groups. Among the most notable results, non-colonized patients showed greater counts of *Enterococcus*, *Ochrobactrum*, and *Staphylococcus (*Figure 2B), whereas a significant increase in the abundance of *Anaerococcus*, *Dialister*, and *Peptoniphilus* was observed in those patients suffering colonization. Interestingly, the analysis of those genera with an abundance lower than 1% also revealed significant differences between non-colonized and colonized patients (Figure 2C). Thus, the non-colonized group presented a higher abundance of *Clostridium* and *Escherichia*, while the colonized group had more *ph2* and *Streptococcus*, although a trend was only obtained with the latter (*p* = 0.085).

### 2.4. Identification of Specific Bacteria as Possible Markers for Predicting MDRB Colonization in SARS-CoV-2 Patients in ICUs

Once it had been established that MDRB colonization in SARS-CoV-2 patients during their stay in the ICU facilities could be associated with several modifications in the relative abundance of some bacteria genera, we further tried to identify some other taxa that could help to predict colonization at the time of entering the ICU. With this aim, ASVs from all samples were analysed to determine core taxa along with the specific bacteria of each group (Figure 3A). Venn diagram analysis showed that both groups of study share 72 core taxa, while only one bacterium (*Jonquetella antrophi*) was identified as specific for the colonized group. Additionally, differential abundance taxa between non-colonized and colonized samples were identified by employing a volcano plot. An increase in the abundance of some bacteria, including *Prevotella timonensis*, *Anaerococcus vaginalis,* and *Dialister propionicifaciens* seems to be associated with MDRB colonization, whereas there was a significant decrease in the abundance of *Ochrobactrum*, *Corynebacterium,* and *Prevotella bivia* in comparison with non-colonized patients (Figure 3B). Finally, ASVs with significantly higher relative abundances in each sample from both groups were determined based on a linear discriminant analysis Effect Size algorithm (LEfSe). Compared by groups, non-colonized patients showed a significantly greater abundance of *Varibaculum cambriense*, *Citrobacter europaeus*, and *Proteus bacterium_R49* members as biomarker taxa, while in the colonized group *Dialister propionicifaciens* was shown as an MDRB biomarker (Figure 3C).

### 2.5. Correlation between Initial Gut Microbiota Composition and Clinical Variables of Non-Colonized and Colonized Patients

The previous results have shown the existence of differences among initial gut microbiota compositions of patients that would end in being colonized or not. To improve the study, we have analysed the correlation between the gut microbiota composition and the clinical variables of the patients. Gut microbiota from non-colonized patients presented only three strong positive correlations (*Porphyromonas*, *Bacteroides uniformis,* and *Prevotella timonensis*) (Figure 4A). In general, for most of the taxa, there was a higher presence of negative correlation with the studied clinical variables. Specifically, the presence of *Corynebacterium* in non-colonized patients was strongly related to lower mortality. On the other hand, the results revealed that stool microbiota from colonized patients was more positively correlated with the clinical variables such as comorbidities, APACHE index, and LDH value (Figure 4B).

## 3. Discussion

Infections caused by MDRB colonization constitute a crucial challenge in patients treated in the ICU [21,22,23]. This problem has been aggravated by the COVID-19 pandemic, in part due to the increase in the number of antibiotic prescriptions in ICUs, being, some of them, unnecessary or not fully justified [24]. In fact, although the data about MDRB colonization in COVID-19 are scant, the empirical treatment with broad spectrum antibiotic therapy and biologics that target and inhibit cytokines, such as IL-1 and IL-6, could raise the risk of MDRB colonization in these patients [25,26]. Our results suggest that demographic factors are independently associated with ICU-acquired MDRB in SASR-CoV-2 patients. The severity index at hospitalization and days in the ICU were also identified as possible independent risk factors for ICU-acquired MDRB in SARS-Cov-2 subjects. Ceccarelli G. et al. stated that risk factors such as comorbidities, mechanical ventilation, and a longer stay in the ICU were responsible for carbapemenase producers infections [27]. Even though a higher number of patients should be evaluated, our results agree with their observations as acquisition of MDRB in the ICU was lightly associated with different comorbidities, symptoms at hospitalization, biochemistry parameters, and mortality rate. These patients also presented a higher necessity for antibiotic treatment. This fact could increase the chances of colonization, as the association between antibiotic treatment and MDRB colonization is well described [28]. Moreover, recent studies have shown that antibiotic treatment in SARS-CoV-2 positive patients was positively correlated to MDRB colonization [29,30]. On the other hand, previous studies have reported that the severity index at hospitalization and days in the ICU are dependent factors for MDRB colonization in SARS-CoV-2 patients compared with virus -free patients [31]. However, the relationship between these factors and SARS-Cov-2 ICU-acquired MDRB has not yet been addressed. Therefore, these findings suggest that the severity and duration of stay at the ICU may be associated with a more frequent use of antibiotics in colonized SARS-CoV-2 patients. Additionally, the dependent factors to colonized status in these patients, including different comorbidities, biochemistry parameters, and mortality rate, might be explained by the reported gut dysbiosis and the inflammatory status in patients with these comorbidities. This previous gut dysbiosis and/or the inflammation condition in these patients may be aggravated with the SARS-CoV-2 infection and facilitate the MDRB colonization. This adds new information for a possible explanation of the MDRB colonization in these COVID patients, however, different studies support our rationale, since it has been described that intestinal dysbiosis in obese patients constitutes a risk factor for SARS-CoV-2 infection, which could lead to severe symptoms with higher MDRB colonization [32].

Furthermore, it is well known that SARS-CoV-2, along with many other virus infections, modifies gut microbiota composition [33]. In this sense, recent reports in SARS-CoV-2 patients have identified an important gut dysbiosis, with enrichment of opportunistic bacterial and fungal pathogens, and depletion of beneficial symbionts that are positively and inversely correlated with SARS-CoV-2 severity, respectively [34,35]. Consequently, it is plausible to hypothesize that gut microbiota can impact MDRB colonization in SARS-CoV-2 patients. Hence, in this pilot study, the results suggest that initial gut microbiota compositions of positive SARS-CoV-2 patients admitted to ICUs might be one of the factors for the development of MDRB colonization during their stay in these facilities. In fact, the SARS-CoV-2 patients colonized by MDRB presented a significant reduction of bacterial diversity. Taking everything in consideration, and even with the limitations of this pilot study, the results suggest that these SARS-CoV-2 patients with a higher gut dysbiosis can be key in the MDRB colonization, together with other key factors such as treatment with broad-spectrum antibiotics. In this scenario, a greater dysbiosis can lead to colonization by pathogenic organisms as well as to an increase the antibiotic resistance gene burden, and subsequent antimicrobial resistance pathogen invasion. Interestingly, previous studies have described a similar impact on positive SARS-CoV-2 patients. Zuo et al. described that SARS-CoV-2 infection increases the presence of opportunistic pathogens, although the virus presence diminished the microbial diversity compared to healthy patients [36].

Closely related to the above, the microbiota composition of colonized SARS-CoV-2 critically ill patients in the ICU facilities used was characterized by an increase of *Anaerococcus*, *Dialister,* and *Peptoniphilus*. Commonly, these bacteria have been recognized as opportunistic pathogens [37], suggesting that the MDRB colonization could derive from the presence and/or the negative effects of these microorganisms in virus-infected patients. Conversely, non-colonized patients presented a higher abundance of *Enterococcus*, *Ochrobactrum,* and *Staphylococcus*. *Enterococci* are lactic acid bacteria comprising both pathogenic and gut symbionts. In fact, many studies have indicated that these microorganisms can produce antimicrobial compounds including bacteriocins [38,39]. Interestingly, our results are in line with the previous studies. Concerning *Ochrobactrum*, they are gram-negative, non-fermenting bacteria classically related to infections both in patients undergoing treatments and subjects outside of a clinical centre with different diseases [40]. However, they have been of low virulence and different studies have indicated that they may be innocuous [41]. Similarly, *staphylococci* have been widely known as pathogenic microorganisms, being *S. aureus* species, a classical penicillin resistant bacterium. However, it has been also published that pre-colonization with *S. aureus* affects the *Pseudomonas aeruginosa* implantation by competitive inhibition [42]. Thus, its presence can be considered as a defensive mechanism. Additionally, other genus taxa such as *Clostridium*, *Escherichia,* and *Corynebacterium* have also been increased in non-colonized SARS-CoV-2 patients. Interestingly, these bacteria have been associated with beneficial effects in human health. For example, some *Clostridium* species have been used as probiotics [43] and as enhancers of the mechanism of action of *Lactobacillus* [44], while for some species of *Corynebacterium* it has been reported that they can produce specific antimicrobial molecules [45]. Briefly, our findings suggest that the presence of these genera of bacteria would point to a high possibility of MDRB colonization and would facilitate a better prognosis for ICU patients infected by SARS-CoV-2.

Aiming to find a precise biomarker for predicting MDRB colonization in SARS-CoV-2 patients, we have determined the existence of the intestinal bacterial species at the time of hospitalization in the ICU of the patients with COVID recruited in our pilot study. Remarkably, the results of gut microbiota of the 8 colonized patients revealed that *Jonquetella atrophy* was exclusively present in the faecal microbiota of colonized patients. This bacterium belongs to the *Synergistetes* phylum, which surprisingly is not usually abundant in the normal microbiota [46]. Nonetheless, this bacterium has been identified in pathologic conditions, and it has been also reported as an opportunistic pathogen [47,48]. The comparison between both groups of study showed that colonized patients presented a significantly higher abundance of *Dialister propionicifaciens*, *Anaerococcus vaginalis,* and *Prevotella timonensis*. These bacteria could be involved in the MDRB colonization in these patients because they have been reported to have antibiotic resistance (*Dialister propionicifaciens*) and have been implicated in inflammatory processes as well as SARS-CoV-2 infections (*Prevotella timonensis* and *Anaerococcus vaginalis*) [49,50,51]. Moreover, the determination of a predictive intestinal microbiome biomarker by LDA plot revealed an enrichment of *Citrobacter europaeus* and *Proteus bacterium*, belonging to *Enterobacterales* order, in non-colonized patients. Specifically, it has been published that *Enterobacteriaceae* members such as *Citrobacter* produce colicins that inhibit the growth of other species [52]. This fact can explain that a higher abundance of these microorganisms in SARS-CoV-2 patients constitutes a protective factor to MDRB colonization. On the contrary, in colonized patients *Dialister propionicifaciens* could be crucial in MDRB colonization. As mentioned above, this bacterium is recognized as an opportunistic pathogen [53] and, moreover, it has been positively correlated with SARS-CoV-2 infection [54,55]. However, its impact on MDRB colonization in these patients remains unknown. For the first time, our results point to this bacterium as a possible and novel predictor MDRB biomarker in SARS-CoV-2 infection. Interestingly, previous metagenomic studies have identified this family as genomes harbouring antibiotic resistance genes [56,57].

Taken together, it has been revealed that determining a predictive biomarker for MDRB in these patients requires a multivariable approach. Therefore, a correlation study between gut microbiota and clinical variables was performed (Figure 4). Our possible non-colonization predictors did not appear as they were not the most abundant species of these patients. However, other mentioned taxa such as *Prevotella bivia*, *Ochrobactrum,* and *Corynebacterium* seemed to be associated with a better prognosis, suggesting again its implication at the time of avoiding MDRB colonization (Figure 4A). On the contrary, a higher abundance of *Dialister*, *Methylobacterium,* and *Porphyromononas* was strongly positively correlated with the development of the MDRB colonization in SARS-CoV-2 patients (Figure 4B). Specifically, the possible biomarker for colonization (*Dialister propionicifaciens*) presented a strong correlation with the presence of comorbidities. As other studies have suggested, an increase in the number of comorbidities at the time of entering the ICU can increase the risk of MDRB colonization [58]. Thus, in the present study, it seems that suffering from certain conditions increases the abundance of *Dialister propiniocifaciens* and, as a result, raises the chances of being colonized.

The findings of this study must be interpreted considering some limitations. Firstly, the number of samples is limited, and although the prevalence of patients colonised by MDRB in our study is high (50%) compared to approximately 5% of patients colonised by other causes, according to our bibliography and our internal data, the limited number of patients means that the results obtained need to be corroborated by a larger study. Another limitation is that it is a single-centre study. Furthermore, it must be taken into account that the patients have received antibiotic treatment, which is one of the most important factors in the production of gut dysbiosis in patients admitted to the ICU.

## 4. Materials and Methods

### 4.1. Patient Population

The study was carried out in the ICU of San Cecilio University Hospital of Granada (Southern Spain) which contains up to 20 individual rooms. Patients aged ≥18 years with an ICU stay >48 h between March and May 2021 and who tested positive for SARS-CoV-2 infection were recruited. All patients were directly admitted to the ICU except for two, which were hospitalized prior to entering the ICU. During the period of study, not all patients could be recruited. In fact, only the first two patients from whom informed consent was obtained each week were recruited. Consequently, the participation rate among all COVID patients admitted to the ICU during the study period was 38.6%.

### 4.2. Ethics Statement

The protocol of this study was approved by the Clinical Research Ethics Committee of Granada (CEIC) (ID of the approval 1133-N-20). All patients gave their consent before being included in the study.

### 4.3. Study Design

This is a cross-sectional pilot study. At the time of entering the ICU, rectal swabs were collected for gut microbiota analysis and for standard surveillance for aerobic Gram-negative bacteria colonization. Colonization detection was carried out by culturing rectal swabs on selective chromogenic media CHROMID^®^ CARBA SMART and CHROMID^®^ ESBL plates (bioMérieux, Marcy-l’Étoile, France). All suspected Gram-negative colonies were analysed by Matrix-Assisted Laser Desorption/Ionization Time-Of-Flight mass spectrometry (MALDI-TOF MS) (Bruker Daltonics, Bremen, Germany) for species identification. Antibiotic resistance phenotypes were determined using the Microscan Walkaway 96 plus system (Beckman Coulter International S.A., Nyon, Switzerland) and observing the guidelines of The European Committee on Antimicrobial Susceptibility Testing (EUCAST) (https://www.eucast.org/eucastguidancedocuments (accessed on 1 July 2022)).

Depending on MDRB colonization results, patients were classified as non-colonized or colonized. As for SARS-CoV-2 infection, main clinical data was registered from these patients: (i) age and gender; (ii) days since symptoms appeared, days in the ICU, and symptoms upon ICU entrance; (iii) biochemical parameters; (iv) clinical severity determined through sequential organ failure assessment (SOFA) and Acute Physiology and Chronic Health disease Classification System II (APACHE II) as well as comorbidities; and (v) care therapies and outcomes.

### 4.4. SARS-CoV-2 Diagnosis

Microbiological diagnosis of SARS-CoV-2 infection was accomplished by detection of SARS-CoV-2 RNA in respiratory samples (oropharyngeal-nasopharyngeal swab, bronchoalveolar lavage, or broncoaspirate), as earlier reported [59]. In brief, total DNA/RNA was extracted from samples by TANBead (Maelstorm 9600, Guadalajara, Spain), and SARS-CoV-2 expression was detected by DIRECT SARS-COV-2 REALTIME PCR KIT with double target: specific for COVID-19 (N gene) and other SARS-related coronaviruses (E gene) regions (Vircell SL, Granada, Spain).

### 4.5. Microbial DNA Extraction, Library Preparation, and Next Generation Sequencing

Faecal DNA was isolated according to the method reported by Rodríguez-Nogales et al. [60]. Total DNA was amplified utilizing primers targeting regions flanking the variable regions 4 through 5 of the bacterial 16 S rRNA gene (V4–5), gel purified, and examined using multiplexing on the Illumina MiSeq machine (Illumina Inc., San Diego, CA, USA). Amplified products were validated visually by running a high-throughput Invitrogen 96-well-E-gel (Thermo Fisher Scientific, Waltham, MA, USA). Then, PCR reactions from the same samples were pooled in one plate, subsequently cleaned, and normalized with the high-throughput Invitrogen SequalPrep 96-well Plate kit. Lastly, the samples were pooled into a library to be fluorometrically measured prior to sequencing. Next-Generation Sequencing (NGS) techniques were performed to sequence the samples using an Illumina MiSeq machine. Raw data for each sample was employed for further analysis of microbiome composition.

### 4.6. Bioinformatics and Statistical Analysis

Bioinformatic analysis of gut microbiota samples was carried out using QIIME2 pipeline (open access, Northern Arizona University, Flagstaff, AZ, USA) [61]. Demultiplexed sequences were loaded into the program and quality control was performed by trimming and filtering, depending on the quality scores of the sequences [62]. Then, denoising was performed by employing DADA2 and amplicons sequence variants (ASVs) were obtained. Taxonomic assignment was calculated against the SILVA reference database [63] and the feature table was filtered to discard both Archaea and Eukaryota features.

Statistical analysis of microbiota data was performed in R. QIIME2 objects were loaded into R to carry out the statistical analysis [64]. The phyloseq package was employed to determine alpha and beta diversity as well as relative abundance. Statistical differences for alpha diversity and relative abundance were analysed using the t-student test when samples followed a normal distribution. When samples did not follow this assumption, a Wilcoxon test was performed. Normality was checked by using the Shapiro–Wilk test included in the Nortest package of R. 

On the other hand, beta diversity differences were calculated with a Permutational Multivariate Analysis of Variance (PERMANOVA) by employing the Adonis function from the Vegan package. Eulerr and MicroViz packages were employed to make the Venn diagram analysis and to represent both heatmaps and correlation plots respectively. Values from correlation plots were calculated with the Pearson coefficient. Finally, the DESeq2 package (version “4.2”) was used to identify differences in taxa expression levels while the microbial package was used to assess possible biomarkers through linear discriminant analysis (LDA) effect size (LEfSe) with an LDA score of 3.

For clinical variables, data was represented as mean ± SD when it followed a normal distribution. In contrast, for non-parametric distributions, median and interquartile range were displayed. For categorical variables, percentages were used.

## 5. Conclusions

The present pilot study provides valuable information on changes in the gut microbiota in critical patients with the SARS-CoV-2 infection developing MDRB colonization. These results, for the first time, suggest that the composition of the intestinal microbiota in patients infected with SARS-CoV-2 at the time of admission to the ICU might be a potential biomarker for MDRB colonization. Specifically, our findings highlighted the role of two genera and one species as possible biomarkers of MDRB in infected subjects: *Anaerococcus*, *Prevotella*, and *Dialister propionicifaciens*. However, in order to further prove these observations, the limitations of this study should be addressed by increasing the number of patients and following the evolution of their gut microbiota along their stay at the ICU.

## Figures and Tables

**Figure 1 antibiotics-12-00498-f001:**
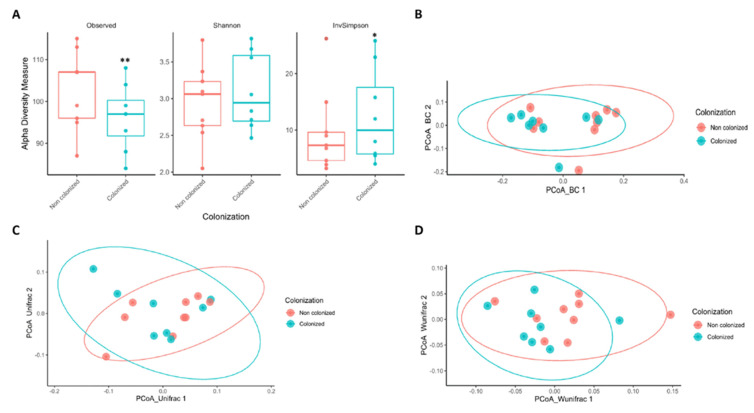
Colonized patients showed differences in gut microbiota richness and evenness compared to non-colonized patients. (**A**) Values of alpha diversity index (Observed features, Shannon, and Inverse Simpson) comparing non-colonized and colonized samples. (**B**) PCoA for Bray–Curtis diversity between non-colonized and colonized samples. (**C**) PCoA for Unifrac distance between non-colonized and colonized samples. (**D**) PCoA for Weighted Unifrac distance between non-colonized and colonized samples. Values are represented as mean ± SD. Significant differences are represented as * = *p* < 0.05; ** = *p* < 0.01. Statistical test employed for alpha diversity analysis was t-test and PERMANOVA test for beta diversity.

**Figure 2 antibiotics-12-00498-f002:**
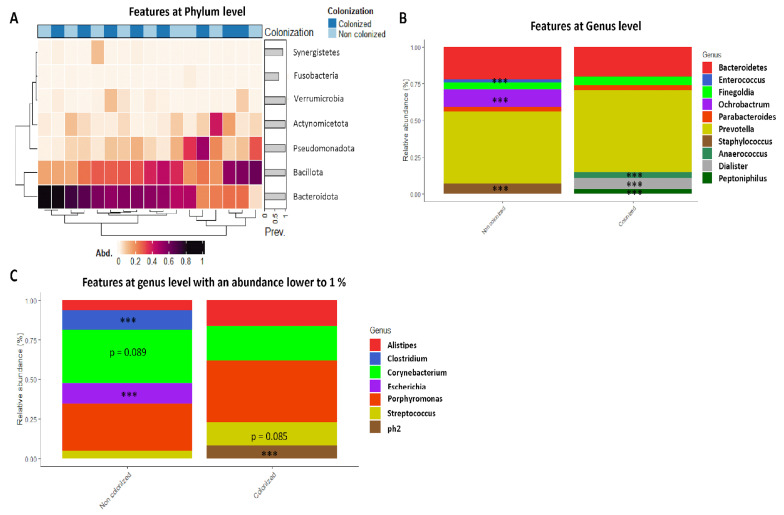
Non-colonized and colonized patients presented differences in some taxa composition at Genus level. (**A**) Heatmap representation of taxa differences based on Phylum annotation. (**B**) Taxa differences at Genus levels for features with an abundance higher than 1%. (**C**) Taxa differences at Genus levels for features with an abundance lower than 1%. Significant differences are represented as *** = *p* < 0.001. Statistical test employed was t-test for parametric distributions and Wilcoxon test for non-parametric distributions.

**Figure 3 antibiotics-12-00498-f003:**
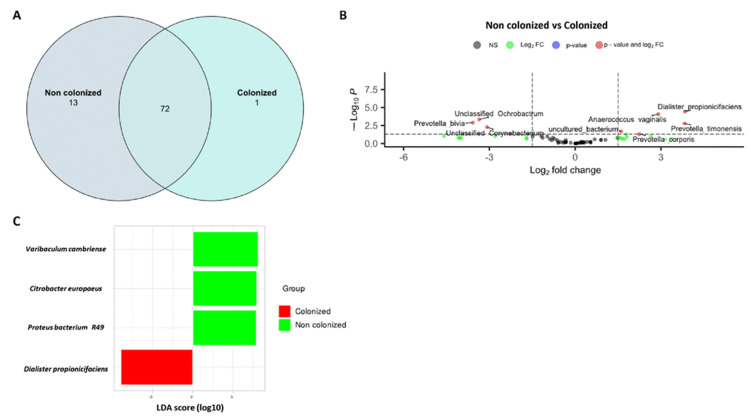
Non-colonized and colonized groups presented differential abundance in some taxa at genus level that could be used as possible biomarkers for predicting colonization. (**A**) Venn diagram showing ASVs distribution between non-colonized and colonized groups (detection level = 0.001 and prevalence = 0.75). (**B**) Volcano plot showing differential abundance taxa of colonized samples vs. non-colonized samples. (**C**) Linear Discriminant Analysis (LDA) Effect Size (LEfSe) plot of taxonomic biomarkers (*p* value = 0.05 and LDA value = 2).

**Figure 4 antibiotics-12-00498-f004:**
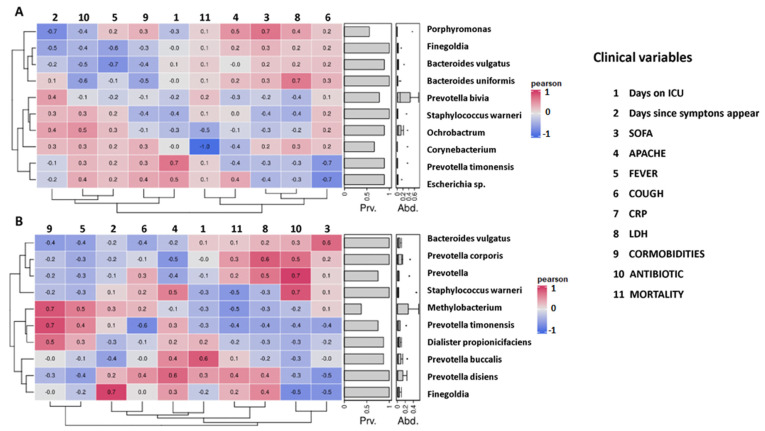
Correlation between gut microbiota of non-colonized and colonized SARS-CoV-2 patients and their clinical variables at the time of entering ICU facilities. (**A**) Correlation plot for non-colonized group. (**B**) Correlation plot for colonized group. The plot only takes into account the most abundant taxa with a prevalence of 90%. Pearson correlation coefficient was calculated for both groups.

**Table 1 antibiotics-12-00498-t001:** Clinical data for recruited patients.

	*N* = 17	
Demographics	Non-Colonized (*n* = 9)	Colonized (*n* = 8)
Age (median, IQR)	64 (58–67)	67 (65–69)
Gender (male)	5 (55%)	6 (75%)
**Comorbidities**		
Obesity (Yes)	2 (22%)	3 (38%)
Pulmonar disease (Yes)	1 (11%)	3 (38%)
Diabetes (Yes)	1 (11%)	3 (38%)
Hypertension (Yes)	3 (33%)	4 (50%)
**Days since symptoms appear (mean ± SD)**	7 ± 1	10 ± 4
**Days in ICU (mean ± SD)**	36 ± 23	31 ± 22
**Symptoms at hospitalization**		
Fever (Yes)	2 (22%)	5 (63%)
Cough (Yes)	2 (22%)	2 (25%)
**Biochemistry parameters**		
LDH (U/L) (mean ± SD)	389 ± 87	559 ± 152
Lymphocytes (%) (median, IQR)	7.8 (5.7–14.2)	7.5 (6.6–8)
CRP (mg/dL) (median, IQR)	87 (26.4–147.8)	93.8 (76.2–152.3)
**Severity index at hospitalization**		
SOFA (median, IQR)	7 (6–8)	7 (6–9)
APACHE (mean ± SD)	15 ± 3	17 ± 4
**Care measures**		
Mechanical ventilation (Yes)	9 (100%)	8 (100%)
Ionotropic support (Yes)	8 (90%)	6 (75%)
Renal replacement therapy (TRR) (Yes)	1 (11%)	2 (25%)
Antibiotic administration (Yes)	4 (44%)	6 (75%)
**Outcome**		
Mortality (Yes)	1 (11%)	3 (38%)

**Table 2 antibiotics-12-00498-t002:** Information of the colonizing bacteria detected in the colonized group. ESBL: extended-spectrum β-lactamase, GNB: Gram-negative Bacilli.

MDRB Colonizing Intestinal Tract	Colonized Patients (*n* = 8)
**ESBL**	
*Escherichia coli*	1 (12 %)
**Carbapenemase producer**	
*Enterobacter cloacae (OXA 48)*	2 (25 %)
*Klebsiella pneumoniae (OXA 48)*	1 (12 %)
*Serratia marcescens (OXA 48)*	1 (12 %)
*Enterobacter cloacae (OXA 48/VIM)*	2 (25 %)
**Non-fermenter GNB**	
*Pseudomonas aeruginosa*	1 (12 %)
*Stenotrophomonas maltophila*	1 (12 %)

## Data Availability

Due to the sensitivity of the data, individual participant data will not be made available. Some data will be available to be shared upon publication by correspondence with either Maria Elena Rodriguez Cabezas (merodri@ugr.es) or Manuel Colmenero (manuel.colmenero.sspa@juntadeandalucia.es), after approval of a proposal, with a signed access agreement, and relevant ethics consent.

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
