# Peer review of "Gut Microbiota Composition Can Predict Colonization by Multidrug-Resistant Bacteria in SARS-CoV-2 Patients in Intensive Care Unit: A Pilot Study"

_antibiotics, 2023, doi:10.3390/antibiotics12030498_

Round 1

Reviewer 1 Report

Excellently written text. The goal of the research is clearly set, methodologically supported. Results presented in a very original way. Discussion with references adequately accompanied the entire article. A job well done

Author Response

Excellently written text. The goal of the research is clearly set, methodologically supported. Results presented in a very original way. Discussion with references adequately accompanied the entire article. A job well done.

Response: We thank the reviewer for the comments made about the manuscript

Reviewer 2 Report

The research is very important. Much research is being done on the association of SARS-CoV-2 infection with the gut microbiota. Gut microbiota is associated with lung pathology and aggravation of the condition of SARS-CoV-2 patients. In manuscript was demonstrated that initial gut microbiota composition of positive SARS-CoV-2 patients admitted to ICU can influence the development of MDRB colonization during their stay in these facilities.

159, Table 2: Change E. clocae to E. cloacae

In the discussion 305-307: on the other side, the microbiota is able to influence the level of cytokines (stimulate / reduce, Including the secretory inhibitor of cytokine) - this is a pathogenetic link. I suggest adding this to the discussion.

Author Response

The research is very important. Much research is being done on the association of SARS-CoV-2 infection with the gut microbiota. Gut microbiota is associated with lung pathology and aggravation of the condition of SARS-CoV-2 patients. In manuscript was demonstrated that initial gut microbiota composition of positive SARS-CoV-2 patients admitted to ICU can influence the development of MDRB colonization during their stay in these facilities.

159, Table 2: Change E. clocae to E. cloacae

Response: Sorry for the misspelling. It has been corrected.

In the discussion 305-307: on the other side, the microbiota is able to influence the level of cytokines (stimulate / reduce, Including the secretory inhibitor of cytokine) - this is a pathogenetic link. I suggest adding this to the discussion.

Response: We thank the reviewer for the comments made about the manuscript, consequently the manuscript has been modified accordingly.

Reviewer 3 Report

This pilot study describes and analyses the gut microbiota composition among 17 SARS-CoV-2 patients in the intensive care unit (ICU) of San Cecilio University Hospital of Granada between March and May 2021. To improve the manuscript, several revisions are required.

Throughout the manuscript:
- Always write in italics the bacterial genus and species (for example, corrected lines 31, 59, 157, 280, 367.

Title:
- Remove the acronym “MDRB” and add “: a pilot study” at the end of the title because it is an important information.

Affiliation:
- “5” in superscript.

Abstract:
- Line 20: remove “ICU” and write “intensive care unit”.
- Line 21: “(MDRB)” should be placed immediately after “multi-drug resistant bacteria”.
- Line 27: “surveillance: Non colonized” use a lower-case letter “n”.
- Line 29: replace the acronym “NGS” by the full name. Moreover, this technique was not presented in the materials and methods section.
- Line 29-30: “Gut microbiota from colonized patients presented lower bacterial diversity compared with non-colonized patients.”: significantly lower or not? Please add p-value in brackets.
- Line 32: “LEfSe” instead of “Lefse”.

Introduction:
- Introduction is well-structured however the aim (lines 75-77) seems not in line with the type of study realized. Indeed, to evaluate an impact, a longitudinal study is required with samples before and after the colonization. But you have realized a cross-sectional study with only one sample at the time of patients was admitted to the ICU. So, a better formulation of your objective seems: “This pilot study aims to compare the gut microbiota among SARS-CoV-2 patients in intensive care unit (ICU) who are colonized or non-colonized by multi-drug-resistant bacteria (MDRB).”

Materials and methods:
- According to instructions of authors of Antibiotics journal, this section should be placed between discussion and conclusion sections.
- All patients responding to inclusion criteria (patients aged ≥ 18 years with an ICU stay > 48 hours between March-May 2021 and tested positive for SARS-CoV-2 infection) were recruited? Or do you exclude some patients?
- Study design: this study was not a case-control study like indicated line 90. In a case-control study researchers study the medical and lifestyle histories of the patients in each group to learn what factors (expositions) may be associated with the disease. You realized a cross-sectional study. Moreover, the term “pilot study” need to be add in this subsection to justify your small patient sample and the absence of a calculation of the number of subjects required in the study.
- Line 99: add a bibliographic reference for the guidelines of the EUCAST.
- Line 103: do you make a difference between “sex and gender” in your study? Why?
- Line 118: add a point “.” After “[22]”.

2.6. Bioinformatics and statistical analysis:
- Justify the text of this paragraph because it is left-aligned.
- Please complete this subsection because several statistical analyses performed and presented in your results are not describe here.
- line 138: it is surprising to use student t-test in your study because your sample is very small (17 patients distributed in two groups of 8 and 9 patients). Non-parametric tests seem more indicated. With which statistical test did you check the normal distribution?

Results:
- The general clinical characteristics of the included patient (lines 185-200) need to be describe first and before the MDRB colonization of patients (lines 152-159).
- Lines 226-229: you indicated “a significant decrease”, can you add the p-values?
- All the results should be presented in the results section: the paragraph lines 404-416 and figure 4 presented results, they should be replaced in the results section (not in discussion).

Tables:
- The citation of table 1 and table 2 are reversed in the text.
- It is not necessary to quote “Table 2” four times in the same paragraph (lines 185-200). The first time is enough.
- The footer of table 1 (EBLSE and GNB) correspond to the footnote of table 2.
- Can you use a word format for tables for a better readability?
- In table 1: regarding APACHE it is mean ± SD not median, IQR.
- Table 1: can you add a p-value column comparing data between non-colonized and colonized groups.
- Table 2: the title is not correct, only colonized patients are concerned.

Figures:
- Figure 3: in the title you describe figure 3 A, B, C and D but they are only A, B and C in the graphs.
- Figure 3B: they are no blue circles (p-values) in the graph as shown in its legend.
- Figure 4: the 4C is not a figure, it is the legend.

Discussion:
- Line 332: “COVID” instead of “covid”.
- Line 336: “[38].” Instead of “.[38]”.
- Why do you realize only a pilot study on a few months with only 17 patients? Is another study planned to complete this pilot study with a larger sample?
- A limitation paragraph is required because the very small size of your patient sample is a big concern. The external validity of the study and the representativeness of the sample need to be discussed (you realized a monocentric study including only 8 colonized patients compared to only 9 non-colonized patients: your results and the generalization of the results are unsure).
- Your discussion section needs to be totally revised because you should use the conditional regarding the interpretation of your results (due to the limitations of your pilot study with 17 patients). Several examples below (not exhaustive).
- Lines 341-343: “in this pilot study, we have demonstrated that initial gut microbiota composition of positive SARS-CoV-2 patients admitted to ICU can influence the development of MDRB colonization during their stay in these facilities.”. This sentence is not true, you cannot demonstrated the initial gut microbiota composition of positive SARS-CoV-2 patients with only 17 patients (it is not a sufficient sample size). Moreover, you cannot make this conclusion from our results because to demonstrate that initial gut microbiota composition can influence the development of MDRB colonization during the ICU stay a longitudinal study is required; however, you only performed a cross-sectional study with only one stool sample at the time of patients was admitted to the ICU.
- Lines 243-244: “In fact, the SARS-CoV-2 patients colonized by MDRB presented a significant reduction of the bacterial diversity.”. Maybe this result is in line with antibiotic treatment because 75% of colonized patients received antibiotics vs only 44% of non-colonized (table 1) and antibiotics are well-known as an important cause of gut dysbiosis.
- Line 375-376: “our findings suggest that the presence of these genera of bacteria could prevent MDRB colonization and would facilitate a better prognosis of ICU patients infected by SARS-CoV-2.”. You cannot use the term prevention because your study is cross-sectional and cannot demonstrate causality, but only a correlation at one time. Regarding the prognostic of ICU patients 1 vs 3 deaths among 9 and 8 patients respectively is not generalizable and they are probably a lots of other explanations for the mortality outcome.
- Lines 380-381: “our results revealed that Jonquetella atrophy was exclusively present in the fecal microbiota of colonized patients.”. Among 8 patients, not among all colonized patients with COVID: you cannot generalize your results with such a small sample.
- Line 400-401: “For the first time, we provided a novel predictor MDRB biomarker in SARS-CoV-2 infection.”. You can only hypothesize that you have found a predictor biomarker but another studies are required with more patients.

Conclusion:
- The conclusion needs to be completely rewritten. For example, your study do not support the sentence line 427-428: “The present pilot study provides a novel approach to prevent MDRB colonization in critical patients with SARS-CoV-2 infection.”.

Funding:
- Correct the mistake line 444: “This research was funded by This work was supported by (…)”.

Author Response

Reviewer 3

This pilot study describes and analyses the gut microbiota composition among 17 SARS-CoV-2 patients in the intensive care unit (ICU) of San Cecilio University Hospital of Granada between March and May 2021. To improve the manuscript, several revisions are required.

Throughout the manuscript:

- Always write in italics the bacterial genus and species (for example, corrected lines 31, 59, 157, 280, 367.

Response: We apologize for the mistakes. The bacterial genus and species have been rewritten in italics.

Title:
- Remove the acronym “MDRB” and add “: a pilot study” at the end of the title because it is important information.

Response: Following the reviewer suggestion, the title has been modified.

Affiliation:
- “5” in superscript.

Response: The number affiliation has been corrected.

Abstract:
- Line 20: remove “ICU” and write “intensive care unit”.
- Line 21: “(MDRB)” should be placed immediately after “multi-drug resistant bacteria”.
- Line 27: “surveillance: Non colonized” use a lower-case letter “n”.
- Line 29: replace the acronym “NGS” by the full name. Moreover, this technique was not presented in the materials and methods section.
- Line 29-30: “Gut microbiota from colonized patients presented lower bacterial diversity compared with non-colonized patients.”: significantly lower or not? Please add p-value in brackets.
- Line 32: “LEfSe” instead of “Lefse”.

Response: Following the reviewer suggestion, the abstract has been modified.

Introduction:
- Introduction is well-structured however the aim (lines 75-77) seems not in line with the type of study realized. Indeed, to evaluate an impact, a longitudinal study is required with samples before and after the colonization. But you have realized a cross-sectional study with only one sample at the time of patients was admitted to the ICU. So, a better formulation of your objective seems: “This pilot study aims to compare the gut microbiota among SARS-CoV-2 patients in intensive care unit (ICU) who are colonized or non-colonized by multi-drug-resistant bacteria (MDRB).”

Response:  We appreciate the comments of the reviewer. As pointed out by the reviewer, we have not performed a longitudinal study, although that is being planned in cooperation with San Cecilio University Hospital. We are trying to obtain samples at different times to gain a better understatement of the relationship between COVID infection/colonization/gut microbiota. Nevertheless, the objective of this study was to identify differences in the initial gut microbiota composition that could be used as possible predictors of colonization. Thus, the manuscript has been modified in order to avoid misunderstanding (Lines 76-78):

“Thus, the present pilot study aims to evaluate whether initial gut microbiota composition of SARS-CoV-2 positive patients in ICU could have an impact in the colonization and establishment of MDRB”.

Materials and methods:

- According to instructions of authors of Antibiotics journal, this section should be placed between discussion and conclusion sections.

Response: We apologize for the mistake. Methods section has been placed between the discussion and conclusions sections.

- All patients responding to inclusion criteria (patients aged ≥ 18 years with an ICU stay > 48 hours between March-May 2021 and tested positive for SARS-CoV-2 infection) were recruited? Or do you exclude some patients?

Response: Not all patients admitted to the ICU during the weeks of the study could be recruited. It was especially troublesome to obtain informed consent or good quality samples in some cases, due to the condition of the patients and the sanitary restrictions. For this reason, the first two patients from whom informed consent was obtained each week were recruited to avoid any recruitment bias.

- Study design: this study was not a case-control study like indicated line 90. In a case-control study researchers study the medical and lifestyle histories of the patients in each group to learn what factors (expositions) may be associated with the disease. You realized a cross-sectional study. Moreover, the term “pilot study” need to be add in this subsection to justify your small patient sample and the absence of a calculation of the number of subjects required in the study.

Response: As it has been suggested, we have modified the type of study performed and we have also included the term “pilot study” so readers understand the low numbers of subjects included (Line 340):

“This is a cross-sectional pilot study.”

- Line 99: add a bibliographic reference for the guidelines of the EUCAST.

Response: We have added the suggested reference.

- Line 103: do you make a difference between “sex and gender” in your study? Why? Response: We are sorry for the mistake. It should say age and gender. We have corrected it in the manuscript.

- Line 118: add a point “.” After “[22]”.

Response: It has been done.

2.6. Bioinformatics and statistical analysis:

- Justify the text of this paragraph because it is left-aligned.

Response: It has been done.

- Please complete this subsection because several statistical analyses performed and presented in your results are not described here.

Response: Thank you for the kind comment. We have added other tests that we mention in the results section, so the information now is more accurate.

-Line 138: it is surprising to use student t-test in your study because your sample is very small (17 patients distributed in two groups of 8 and 9 patients). Non-parametric tests seem more indicated. With which statistical test did you check the normal distribution?

-Response: We thank again to the reviewer for the comment, and it is an interesting suggestion. We know that our sample size is small, but the minimum size for evaluating normality is usually 5. Thus, we checked normality by employing Shapiro Wilk test and depending on the result, we used t-test or Wilcoxon test. Here you can find the results of normality assumption:

Results:
- The general clinical characteristics of the included patient (lines 185-200) need to be described first and before the MDRB colonization of patients (lines 152-159).

-Response: According the reviewer suggestions the manuscript has been modified.  

- Lines 226-229: you indicated “a significant decrease”, can you add the p-values?

Response: Following the reviewer suggestions, the p-values have been added.

- All the results should be presented in the results section: the paragraph lines 404-416 and figure 4 presented results, they should be replaced in the results section (not in discussion).

Response: As the reviewer has suggested, we thought that he is right, and we have relocated figure 4 to the results section.

Tables:
- The citation of table 1 and table 2 are reversed in the text.

Response: We are sorry for the mistake, and it has been modified accordingly

- It is not necessary to quote “Table 2” four times in the same paragraph (lines 185-200). The first time is enough.

Response: It has been done

- The footer of table 1 (EBLSE and GNB) correspond to the footnote of table 2.

Response: We are sorry for the mistake. It has been corrected.

- Can you use a word format for tables for a better readability?

Response: the table format has been modified.

- In table 1: regarding APACHE it is mean ± SD not median, IQR.

Response: We apologize for the mistake. It has been corrected.

- Table 1: can you add a p-value column comparing data between non-colonized and colonized groups.

Response: We kindly appreciate your suggestion. However, as we have mentioned within the text, we only found one significant difference for all the variables (LDH: p<0.05). As a result, we did not include the p values because we thought it was not going to be useful information.

- Table 2: the title is not correct, only colonized patients are concerned.

Response: It has been fixed.

Figures:
- Figure 3: in the title you describe figure 3 A, B, C and D but they are only A, B and C in the graphs.

Response: As mentioned by the reviewer, there was a mistake in the caption of figure 3. Therefore, the figure 3 has been corrected.

- Figure 3B: they are no blue circles (p-values) in the graph as shown in its legend.

Response: We appreciate the observation made by the reviewer about this issue. Legend of figure 3B (volcano plot) is displayed by default. P-value as blue circle appears in case there would be a point between vertical lines as is the region defined only by p-value.

- Figure 4: the 4C is not a figure, it is the legend.

Response: As suggested by the reviewer, we have erased 4C note as it is the legend of figure 4.

Discussion:
- Line 332: “COVID” instead of “covid”.

Response: It has been changed.

- Line 336: “[38].” Instead of “.[38]”.

Response: It has been corrected.

- Why do you realize only a pilot study on a few months with only 17 patients? Is another study planned to complete this pilot study with a larger sample?

Response: Although the selection criteria for this pilot study were broad, not all patients admitted to the ICU during the weeks of the study could be recruited. Particularly problematic was obtaining informed consent, or the possibility of obtaining good quality samples. For this reason, the first two patients from whom informed consent was obtained each week were recruited to avoid any recruitment bias.

Cross-sectional studies of this type are useful for establishing preliminary evidence for planning a future advanced study. In our case, the results obtained will help us to carry out future larger studies in which to study the gut microbiota not only in patients infected with COVID-19, but also in other patients requiring antibiotic treatment during their stay in ICU.

- A limitation paragraph is required because the very small size of your patient sample is a big concern. The external validity of the study and the representativeness of the sample need to be discussed (you realized a monocentric study including only 8 colonized patients compared to only 9 non-colonized patients: your results and the generalization of the results are unsure).

Response: The findings of this study must be interpreted considering some limitations. Firstly, the number of samples is limited, and although the prevalence of patients colonized by MDRO in our study is high (50%) compared to approximately 5% of patients colonized by other causes according to bibliography and our internal data, the limited number of patients means that the results obtained need to be corroborated by a larger study. Another limitation is that it is a single-center study. Furthermore, it must be considered that the patients have received antibiotic treatment, which is one of the most important factors in the production of gut dysbiosis in patients admitted to the ICU.

- Your discussion section needs to be totally revised because you should use the conditional regarding the interpretation of your results (due to the limitations of your pilot study with 17 patients). Several examples below (not exhaustive).

Response: The discussion section has been thoroughly revised with a conditional approach considering the limitations of the study.

- Lines 341-343: “in this pilot study, we have demonstrated that initial gut microbiota composition of positive SARS-CoV-2 patients admitted to ICU can influence the development of MDRB colonization during their stay in these facilities.”. This sentence is not true, you cannot demonstrate the initial gut microbiota composition of positive SARS-CoV-2 patients with only 17 patients (it is not a sufficient sample size). Moreover, you cannot make this conclusion from our results because to demonstrate that initial gut microbiota composition can influence the development of MDRB colonization during the ICU stay a longitudinal study is required; however, you only performed a cross-sectional study with only one stool sample at the time of patients was admitted to the ICU.

Response: This sentence has been rephrased with the reviewer comments and issues.

- Lines 243-244: “In fact, the SARS-CoV-2 patients colonized by MDRB presented a significant reduction of the bacterial diversity.”. Maybe this result is in line with antibiotic treatment because 75% of colonized patients received antibiotics vs only 44% of non-colonized (table 1) and antibiotics are well-known as an important cause of gut dysbiosis.

Response: It has been rephrased according to the reviewer’s comments and concerns.

- Line 375-376: “our findings suggest that the presence of these genera of bacteria could prevent MDRB colonization and would facilitate a better prognosis of ICU patients infected by SARS-CoV-2.”. You cannot use the term prevention because your study is cross-sectional and cannot demonstrate causality, but only a correlation at one time. Regarding the prognostic of ICU patients 1 vs 3 deaths among 9 and 8 patients respectively is not generalizable and they are probably a lots of other explanations for the mortality outcome.

Response: We appreciate the reviewer’s comments. Accordingly, we have rephrased this sentence without the term “prevention” and we have modified the manuscript considering the limitations of the study and the conclusions we can draw.

- Lines 380-381: “our results revealed that Jonquetella atrophy was exclusively present in the fecal microbiota of colonized patients.”. Among 8 patients, not among all colonized patients with COVID: you cannot generalize your results with such a small sample.

Response: According the reviewer comment, we have rewritten this sentence and make a remark in the paragraph indicating the limitations of the study at the end of this discussion section.

- Line 400-401: “For the first time, we provided a novel predictor MDRB biomarker in SARS-CoV-2 infection.”. You can only hypothesize that you have found a predictor biomarker but another studies are required with more patients.

Response: We have rewritten this sentence as follows:

“For the first time, our results point to this bacterium as a possible and novel predictor MDRB biomarker in SARS-CoV-2 infection. Interestingly, previous metagenomic studies have identified this family as genomes harbouring antibiotic resistance genes [61,62].”

Conclusion:
- The conclusion needs to be completely rewritten. For example, your study do not support the sentence line 427-428: “The present pilot study provides a novel approach to prevent MDRB colonization in critical patients with SARS-CoV-2 infection.”.

Response: The conclusion section has been completely rewritten.

Funding:
- Correct the mistake line 444: “This research was funded by This work was supported by (…)”.

Response: We apologize for the mistake. The manuscript has been modified accordingly.

Round 2

Reviewer 3 Report

Dear authors,

Thank you for your responses and congratulations on the revisions and improvement made to your manuscript.

For a better understanding of readers, I think you should complete your “4.1. Patient population” subsection to explain the sampling procedure to include patients as explained in your response letter: not all patients admitted to the ICU during the weeks of the study could be recruited, only the first two patients from whom informed consent was obtained each week were recruited. You could also add the participation/inclusion rate among all COVID patients admitted to ICU during your study period.

Add the acronym “(NGS)” in the title of the subsection 4.5 after “next generation sequencing” because this acronym was used after without previous definition.

Congratulations on your research and good luck for the future.

Author Response

We  highly appreciate the reviewer helpful and insightful comments. Following the reviewer comments, we have modified the manuscript.